# Trehalose-Based Nucleolipids as Nanocarriers for Autophagy Modulation: An In Vitro Study

**DOI:** 10.3390/pharmaceutics14040857

**Published:** 2022-04-13

**Authors:** Anthony Cunha, Alexandra Gaubert, Julien Verget, Marie-Laure Thiolat, Philippe Barthélémy, Laurent Latxague, Benjamin Dehay

**Affiliations:** 1Univ. Bordeaux, INSERM U1212, CNRS UMR 5320, ARNA, ARN: Régulations Naturelle et Artificielle, ChemBioPharm, 146 rue Léo Saignat, F-33076 Bordeaux, France; antcunha@u-bordeaux.fr (A.C.); alexandra.gaubert@u-bordeaux.fr (A.G.); julien.verget@u-bordeaux.fr (J.V.); philippe.barthelemy@inserm.fr (P.B.); laurent.latxague@u-bordeaux.fr (L.L.); 2Univ. Bordeaux, CNRS, IMN, UMR 5293, F-33000 Bordeaux, France; marie-laure.thiolat@u-bordeaux.fr

**Keywords:** nucleolipids, nanoparticles, PLGA, trehalose, autophagy

## Abstract

The Autophagy Lysosomal Pathway is one of the most important mechanisms for removing dysfunctional cellular components. Increasing evidence suggests that alterations in this pathway play a pathogenic role in Parkinson’s disease, making it a point of particular vulnerability. Numerous studies have proposed nanotechnologies as a promising approach for delivering active substances within the central nervous system to treat and diagnose neurodegenerative diseases. In this context, the aim was to propose the development of a new pharmaceutical technology for the treatment of neurodegenerative diseases. We designed a trehalose-based nanosystem by combining both a small natural autophagy enhancer molecule named trehalose and an amphiphilic nucleolipid conjugate. To improve nucleolipid protection and cellular uptake, these conjugates were formulated by rapid mixing in either solid lipid nanoparticles (Ø = 120.4 ± 1.4 nm) or incorporated into poly(lactic-co-glycolic acid) nanoparticles (Ø = 167.2 ± 2.4 nm). In vitro biological assays demonstrated a safe and an efficient cellular uptake associated with autophagy induction. Overall, these nucleolipid-based formulations represent a promising new pharmaceutical tool to deliver trehalose and restore the autophagy impaired function.

## 1. Introduction

Due to their prevalence, high societal costs, and the absence of disease-modifying or neuroprotective therapeutic strategies, one of the major challenges today lies in discovering new therapies for neurodegenerative diseases (NDD). Parkinson’s disease (PD) is the second most common NDD after Alzheimer’s disease [1]. PD is characterized by a selective neuronal vulnerability, including degeneration in specific brain regions hosting dopaminergic neurons and deposits of misfolded α-synuclein-rich protein aggregates [2]. Among the pathological mechanisms involved in the pathogenesis of PD, alteration of the lysosomal function of dopaminergic neurons is of increasing interest in PD research. Lysosomal impairment is now described as a major contributor to the pathogenesis of this disease [3]. Indeed, lysosomes are organelles responsible for the clearance, through autophagy, of long-lived proteins, such as α-synuclein, which tend to aggregate due to their misfolding. In PD, α-synuclein aggregation interferes with cellular homeostasis and impairs cellular degradation systems, leading to neuronal death. Thus, restoring lysosomal function represents one of the attractive pathological targets for developing new therapeutic approaches. Trehalose belongs to the therapeutic arsenal against lysosomal failure. Trehalose is a non-reducing disaccharide widely used in the food industry [4], which has recently shown unique properties, particularly in preventing neurodegeneration. Trehalose can act as a potent protein stabilizer, preserving the structural integrity of proteins and inducing autophagy to improve the clearance of toxic proteins [5,6,7]. Trehalose treatment reduced the level of toxic protein aggregates, thereby improving behavioural symptoms and survival in animal models of neurodegenerative diseases, including Alzheimer’s [8], Parkinson’s [9,10], and Huntington’s diseases [11]. However, this small molecule is widely distributed upon administration and needs to be specifically delivered to the damaged brain neurons.

One approach to overcoming this limitation relies on the use of nanotechnologies [12,13] such as poly(lactic-co-glycolic acid) nanoparticles (PLGA NPs) [14,15] and solid lipid nanoparticles (SLNs) based on nucleolipids (NLs) [16,17]. First, PLGA NPs present many advantages for their use in the biomedical field due to their biocompatibility, improved drug solubility, efficient targeting, better cellular internalization, … [18]. Previous studies showed that empty PLGA NPs restored lysosome acidification failures, particularly in experimental models of PD [19,20,21,22]. Second, NL-based SLNs also represent good candidates [23] due to their bio-inspired nature, biocompatibility, biological activities (antimicrobial properties, antifungal, antiviral, antitumor, …), and their remarkable self-assembly capacity [24,25]. Indeed, these amphiphilic molecules can form supramolecular objects such as SLNs, micelles, or liposomes, which can be used to carry active substances, such as deoxyribonucleic acid (DNA), antisense oligonucleotides or small interfering ribonucleic acid (siRNA) directly into cells [26]. Moreover, thanks to molecular recognition, nucleosides can pass cellular membranes through specific transporters, while their lipidization allows their absorption by passive diffusion [27,28]. In addition, we have recently demonstrated that the latter was a good strategy for facilitating passage across the blood brain barrier (BBB), as in the case of uridine with a lipidic diketal (C15), used as a neurotracer, which successfully crossed the BBB [29]. Despite the many advantages of nanotechnology for the delivery of active substances, only a few studies have focused on the delivery of trehalose using nanovectors for the treatment of neurodegenerative diseases, including PD [30,31,32].

In this study, we report the synthesis of trehalose-NLs (hereafter called **GNLs** for glyco-nucleolipids) embedded in PLGA NPs and SLNs as original nanosystems. We demonstrated in vitro that the combination of trehalose and NL did not induce any cellular toxicity, while it improved the uptake and internalization of **GNLs** into neuronal cells associated with the trehalose-induced autophagy effect. The encouraging results described herein suggest that **GNL**-based nanovectors may be efficient pharmaceutical nanocarriers for trehalose delivery.

## 2. Materials and Methods

### 2.1. Procedure for Glyconucleolipids Synthesis

All reactions were performed under an argon atmosphere. Unless otherwise stated, yields were determined on chromatographically and spectroscopically (nuclear magnetic resonance (^1^H-NMR)) homogeneous materials. All reagent-grade chemicals were purchased from commercial suppliers and used as received, unless otherwise stated. ^1^H-NMR and ^13^C-NMR spectra were acquired at 293 K (unless otherwise indicated) on a Bruker Avance 300 (^1^H: 300 MHz, ^13^C: 75.46 MHz) spectrometer with residual CHCl_3_ used as internal reference (7.26 ppm). The chemical shifts (δ) and coupling constants (J) were indicated in ppm and Hz, respectively. The following abbreviations were applied to explain the multiplicities: s = singlet, d = doublet, t = triplet, q = quartet, m = multiplet, b = broad. Fourier Transform InfraRed (FT-IR) spectra were acquired on a Perkin-Elmer spectrometer Spectrum two (UATR two). A Waters Micromass ZQ instrument equipped with an electrospray source (positive and/or negative mode) was used for the electrospray ionization high-resolution mass spectrometry (ESI HRMS) analyses. Matrix-assisted laser desorption ionization—time of flight (MALDI TOF) mass spectrometric analyses, using 3,4-dihydroxybenzoic acid as the matrix, were performed on a PerSeptive Biosystems Voyager-De Pro MALDI mass spectrometer in the Linear mode. Analytical thin-layer chromatography was performed using silica gel 60 F254 pre-coated plates (Merck Millipore, Saint Quentin Fallavier, France) with a revelation by UV light, potassium permanganate or sulfuric acid solutions. Silica gel (0.043–0.063 mm) was used for flash chromatography.

### 2.2. Synthesis of Compound ***1***: 5′-Azido-5′-deoxythymidine

Triphenylphosphine (1.2 equiv., 2.6 g, 9.91 mmol), sodium azide (5 equiv., 2.68 g, 41.3 mmol) and carbon tetrabromide (1.2 equiv., 3.29 g, 9.91 mmol) were sequentially added to a solution of thymidine (1 equiv., 2 g, 8.26 mmol), in anhydrous dimethylformamide (DMF) (40 mL) and under argon. The mixture was stirred for 24 h at room temperature until completion of the reaction. The reaction was then quenched by the addition of an aq. NaHCO_3_ solution (50 mL) and the resulting mixture diluted with water (30 mL). Three extractions with dichloromethane (DCM; 3 × 30 mL) were applied to the aqueous phase, before drying the combined organic phases over Na_2_SO_4_ and concentrating to dryness under vacuum. A purification of the resulting crude by flash chromatography on a silica gel (DCM/MeOH, gradient 99:1 to 95:5) was required to obtain the expected compound **1** as a white powder (1.63 g, 74%). NMR data were consistent with the literature [33]. **R_f_:** 0.37 (DCM/MeOH, 95:5). **^1^H NMR** (300 MHz, MeOD) δ (ppm): 1.90 (d, *J* = 0.9 Hz, 3H, CH_3_ thymidine), 2.25–2.33 (m, 2H, H_2′_), 3.56 (dd, *J* = 3.9 Hz, 13.2 Hz, 1H, H_5a′_), 3.66 (dd, *J* = 5.1 Hz, 13.5 Hz, 1H, H_5b′_), 3.96 (q, *J* = 3.9 Hz, 9 Hz, 2H, H_4′_), 4.32–4.37 (m, 1H, H_3′_), 6.27 (t, *J* = 6.9 Hz, 1H, H_1′_), 7.54 (s, 1H, H_6_).

### 2.3. Synthesis of Compound ***2***: 3-(5′-Azido-5′-deoxythymidin-3-yl) Methyl Propanoate

DBU (diazabicycloundecene) (1.55 equiv., 433 µL, 2.90 mmol) and methyl acrylate (1.55 equiv., 261 µL, 2.90 mmol) were sequentially added to a solution of **1** (1 equiv., 0.500 g, 1.87 mmol) in anhydrous DMF (2 mL) and under argon. The mixture was stirred overnight at 60 °C until the completion of the reaction and followed by concentration to dryness under vacuum. The resulting crude was purified by flash chromatography on a silica gel (DCM/MeOH, gradient 98:2 to 96:4) to obtain the expected compound **2** as a colorless oil (0.436 g, 66%). **R_f_:** 0.43 (DCM/MeOH, 95:5). **^1^H NMR** (300 MHz, CDCl_3_) δ (ppm): 1.89 (d, *J* = 0.9 Hz, 3H, CH_3_ thymidine), 2.09–2.23 (m, 1H, H_2′a_), 2.28–2.41 (m, 1H, H_2′b_), 2.59 (t, *J* = 7.5 Hz, 2H, H_B_ linker), 3.55 (dd, *J* = 3.6 Hz, 13.2 Hz, 1H, H_5a′_), 3.68 (dd, *J* = 3.0 Hz, 12.9 Hz, 1H, H_5b′_), 3.62 (s, 3H, H_C_ linker), 4.01 (q, *J* = 3.6 Hz, 1H, H_4′_), 4.18 (t, *J* = 7.8 Hz, 2H, H_A_ linker), 4.33–4.43 (m, 1H, H_3′_), 6.26 (t, *J* = 6.9 Hz, 1H, H_1′_), 7.34 (s, 1H, H_6_). **^13^C NMR** (75.46 MHz, CDCl_3_) δ (ppm): 13.3 (CH_3_ thymidine), 32.1 (CH_2_-B linker), 37.2 (CH_2_-A linker), 40.2 (C_2′_), 51.9 (CH_3_-C linker), 52.2 (C_5′_), 71.4 (C_3′_), 84.5 (C_4′_), 85.5 (C_1′_), 110.4 (C_5_), 133.9 (C_6_), 150.6 (C=O thymidine), 163.2 (C=O thymidine), 171.9 (C=O linker).

### 2.4. Synthesis of Compound ***3***: 3-[3′-O-(tert-butyldimethylsilyl)-5′-azido-5′-deoxythymidin-3-yl] Methyl Propanoate

TBDMSCl (*tert*-Butyldimethylsilyl chloride) (1.2 equiv., 0.220 g, 1.46 mmol) and imidazole (1.2 equiv., 0.099 g, 1.46 mmol) were sequentially added to a solution of **2** (1 equiv., 0.430 mg, 1.22 mmol) in anhydrous DMF (2.6 mL) and under argon. The mixture was stirred overnight at room temperature until the completion of the reaction. DCM was added and the precipitate formed was filtered. The mixture was then diluted with toluene (100 mL), and DMF was co-evaporated under vacuum to obtain a yellow oil. The resulting crude was purified by flash chromatography on a silica gel (petroleum ether/EtOAc, 70:30) to obtain the expected compound **3** as a colorless oil (0.491 g, 86%). **R_f_:** 0.63 (petroleum ether/EtOAc, 70:30). **^1^H NMR** (300 MHz, CDCl_3_) δ (ppm): −0.05 (s, 6H, CH_3_-Si), 0.75 (s, 9H, ^t^Bu), 1.80 (s, 3H, CH_3_ thymidine), 1.97–2.24 (m, 2H, H_2′_), 2.49 (t, *J* = 7.5 Hz, 2H, H_B_ linker), 3.38 (dd, *J* = 3.9 Hz, 13.5 Hz, 1H, H_5a′_), 3.48–3.59 (m, 4H, H_C_ linker et H_5b′_), 3.78–3.85 (m, 1H, H_4′_), 4.09 (t, *J* = 7.8Hz, 2H, H_A_ linker), 4.19–4.27 (m, 1H, H_3′_), 6.13 (t, *J* = 6.6 Hz, 1H, H_1′_), 7.20 (s, 1H, H_6_). **^13^C NMR** (75.46 MHz, CDCl_3_) δ (ppm): −5.17 (CH_3_-Si), −4.9 (CH_3_-Si), 13.0 (CH_3_ thymidine), 17.6 (Cq tBu), 25.4 (CH_3_ ^t^Bu), 31.8 (CH_2_-B linker), 36.8 (CH_2_-A linker), 40.4 (C_2′_), 51.4 (CH_3_-C linker), 51.5 (C_5′_), 71.5 (C_3′_), 84.6 (C_4′_), 85.3 (C_1′_), 110.0 (C_5_), 133.4 (C_6_), 150.3 (C=O thymidine), 162.7 (C=O thymidine), 171.3 (C=O linker).

### 2.5. Synthesis of Compound ***4***: Propargyl N-dodecylcarbamate

Triethylamine (1.05 equiv., 0.229 g, 2.27 mmol) and drop by drop propargyl chloroformate (1 equiv., 0.256 g, 2.16 mmol) were sequentially added to a solution of dodecylamine (1 equiv., 0.400 g, 2.16 mmol) in anhydrous DCM (6 mL) and under argon. The mixture was stirred for 3 h at 0 °C until the completion of the reaction, followed by concentration to dryness under vacuum. The resulting crude was purified by flash chromatography on a silica gel (petroleum ether/EtOAc, 90:10) to obtain the expected compound **4** as a colorless oil (0.333 g, 97%). **R_f_:** 0.61 (petroleum ether/EtOAc, 90:10). **^1^H NMR** (300 MHz, CDCl_3_) δ (ppm): 0.72–0.85 (m, 3H, CH_3_ lipid), 1.19 (s, 18H, CH_2_ lipid), 1.37–1.52 (m, 2H, CH_2_-CH_3_ lipid), 2.40 (s, 1H, CH propargyl), 3.03–3.17 (m, 2H, CH_2_-NH lipid), 4.60 (s, 2H, CH_2_ propargyl), 4.99–5.24 (bs, 1H, NH lipid). **^13^C NMR** (75.46 MHz, CDCl_3_) δ (ppm): 14.1 (CH_3_ lipid), 22.7 (CH_2_ lipid), 26.7 (CH_2_ lipid), 29.3 (CH_2_ lipid), 29.3 (CH_2_ lipid), 29.5 (CH_2_ lipid), 29.6 (CH_2_ lipid), 29.6 (CH_2_ lipid), 29.6 (CH_2_ lipid), 29.8 (CH_2_-CH_3_ lipid), 31.9 (CH_2_ lipid), 41.2 (CH_2_-NH lipid), 52.2 (CH_2_ propargyl), 74.4 (CH propargyl), 78.4 (C propargyl), 155.5 (C=O lipid).

### 2.6. Synthesis of Compound ***5***: 3′-[3′-O-(tert-butyldimethylsilyl)-5′-{4-[(dodecylcarbamoyloxy)methyl]-1H-1,2,3-triazol-1-yl}-5′-deoxythymidin-3-yl] Methyl Propanoate

Compound **4** (1 equiv., 0.181 g, 0.68 mmol), copper (II) sulfate pentahydrate (0.1 equiv., 0.011 g, 0.07 mmol) and sodium ascorbate (0.2 equiv., 0.027 g, 0.14 mmol) were sequentially added to a solution of **3** (1 equiv., 0.316 g, 0.68 mmol) in a mixture of H_2_O/THF (tetrahydrofuran) (1:1, 5 mL). The mixture was stirred overnight at 60 °C until the completion of the reaction, followed by concentration to dryness under vacuum to obtain a green product. This was washed several times with EDTA (Ethylenediaminetetraacetic acid) until colorless organic phases were obtained, followed by concentration to dryness under vacuum. The resulting crude was purified by flash chromatography on a silica gel (toluene/acetone, 80:20) to obtain the expected compound **5** as a white powder (0.433 g, 87%). **R_f_:** 0.33 (toluene/acetone, 80:20). **^1^H NMR** (300 MHz, CDCl_3_) δ (ppm): 0.04–0.07 (m, 6H, CH_3_-Si), 0.78–0.92 (m, 12H, CH_3_ lipid et ^t^Bu), 1.20 (m, 18H, CH_2_ lipid), 1.33–1.49 (m, 2H, CH_2_-CH_3_ lipid), 1.84 (s, 3H, CH_3_ thymidine), 2.00–2.77 (m, 2H, H_2′_), 2.58 (t, *J* = 7.2 Hz, 2H, H_B_ linker), 3.03–3.14 (m, 2H, CH_2_-NH lipid), 3.62 (s, 3H, H_C_ linker), 4.02–4.11 (m, 1H, H_4′_), 4.17 (t, *J* = 7.8 Hz, 2H, H_A_ linker), 4.34–4.48 (m, 1H, H_3′_), 4.52–4.63 (m, 2H, H_5′_), 4.93–5.04 (m, 1H, NH lipid), 5.10 (s, 2H, O-CH_2_ triazole), 6.15 (t, *J* = 6.6 Hz, 1H, H_1′_), 6.60 (s, 1H, H_6_), 7.66 (s, 1H, CH triazole). **^13^C NMR** (75.46 MHz, CDCl_3_) δ (ppm): −4.8 (CH_3_-Si), −4.6 (CH_3_-Si), 13.2 (CH_3_ thymidine), 14.1 (CH_3_ lipid), 17.8 (Cq ^t^Bu), 22.7 (CH_2_ lipid), 25.7 (CH_3_ ^t^Bu), 26.7 (CH_2_ lipid), 29.3 (CH_2_ lipid), 29.3 (CH_2_ lipid), 29.6 (CH_2_ lipid), 29.6 (CH_2_ lipid), 29.6 (CH_2_ lipid), 29.8 (CH_2_-CH_3_ lipid), 29.9 (CH_2_ lipid), 31.9 (CH_2_ lipid), 32.1 (CH_2_-B linker), 37.1 (CH_2_-A linker), 39.5 (C_2′_), 41.1 (CH_2_-NH lipid), 50.5 (C_5′_), 51.8 (CH_3_-C linker), 57.7 (O-CH_2_ triazole), 71.7 (C_3′_), 83.9 (C_4′_), 86.0 (C_1′_), 110.8 (C_5_), 125.6 (CH triazole), 133.8 (C_6_), 143.6 (C triazole), 150.5 (C=O thymidine), 156.2 (C=O lipid), 162.9 (C=O thymidine), 171.6 (C=O linker). **HRMS** (ESI) [M + H^+^]: calcd= 735.44712, found= 735.44576.

### 2.7. Synthesis of Compound ***6***: 3′-[3′-O-(tert-butyldimethylsilyl)-5′-{4-[(dodecylcarbamoyloxy) methyl]-1H-1,2,3-triazol-1-yl}-5′-deoxythymidin-3-yl] Propanoic Acid

Compound **5** (1 equiv., 0.431 g, 0.59 mmol) was dissolved in 1,4-dioxane (13 mL) before adding a solution of NaOH (1N) in MeOH (1.3 mL). The mixture was stirred overnight at room temperature until the completion of the reaction. The reaction mixture was treated with a solution of HCl (5%) to reach neutrality and then concentrated to dryness under vacuum. The resulting crude was purified by flash chromatography on a silica gel (DCM/MeOH, 90:10) to obtain the expected compound **6** (0.310 g, 73%). **R_f_:** 0.26 (DCM/MeOH, 90:10). **^1^H NMR** (300 MHz, CDCl_3_) δ (ppm): −0.05–0.05 (m, 6H, CH_3_-Si), 0.73–0.85 (m, 12H, CH_3_ lipid et ^t^Bu), 1.10–1.27 (m, 18H, CH_2_ lipid), 1.29–1.47 (m, 2H, CH_2_-CH_3_ lipid), 1.79 (s, 3H, CH_3_ thymidine), 2.07–2.21 (m, 2H, H_2′_), 2.37–2.51 (m, 2H, H_B_ linker), 2.89–3.11 (m, 2H, CH_2_-NH lipid), 3.97–4.15 (m, 3H, H_4′_ et H_A_ linker), 4.30–4.43 (m, 1H, H_3′_), 4.50–4.59 (m, 2H, H_5′_), 5.07 (s, 2H, O-CH_2_ triazole), 5.21–5.31 (m, 1H, NH lipid), 6.09 (t, *J* = 6.6 Hz, 1H, H_1′_), 6.61 (s, 1H, H_6_), 7.67 (s, 1H, CH triazole), 7.75–8.12 (bs, 1H, COOH linker). **^13^C NMR** (75.46 MHz, CDCl_3_) δ (ppm): −4.9 (CH_3_-Si), −4.8 (CH_3_-Si), 13.1 (CH_3_ thymidine), 14.0 (CH_3_ lipid), 17.7 (Cq tBu), 22.6 (CH_2_ lipid), 25.5 (CH_3_ ^t^Bu), 26.7 (CH_2_ lipid), 29.2 (CH_2_ lipid), 29.2 (CH_2_ lipid), 29.3 (CH_2_ lipid), 29.4 (CH_2_ lipid), 29.5 (CH_2_ lipid), 29.5 (CH_2_-CH_3_ lipid), 29.8 (CH_2_ lipid), 31.8 (CH_2_ lipid), 33.6 (CH_2_-B linker), 38.0 (CH_2_-A linker), 39.2 (C_2′_), 41.0 (CH_2_-NH lipid), 50.6 (C_5′_), 57.6 (O-CH_2_ triazole), 71.8 (C_3′_), 84.0 (C_4′_), 86.3 (C_1′_), 110.5 (C_5_), 125.3 (CH triazole), 133.8 (C_6_), 143.5 (C triazole), 150.4 (C=O thymidine), 156.1 (C=O lipid), 162.9 (C=O thymidine), 175.9 (C=O linker). **HRMS** (ESI) [M + H^+^]: calcd = 721.43147, found = 721.43198.

### 2.8. Synthesis of Compound ***7***: 2, 2′, 3, 3′, 4, 4′, 6′-Hepta-O- (trimethylsilyl)-α,α’-D-trehalose

D-(+)-trehalose dihydrate (1 equiv., 0.500 g, 1.32 mmol) was co-evaporated to dryness under vacuum with anhydrous DMF (3 × 10 mL). Following this, *N,O*-bistrimethylsilylacetamide (8.6 equiv., 2.78 mL, 11.37 mmol) and TBAF (Tetra-n-butylammonium fluoride) (0.06 equiv., 79 µL, 0.08 mmol) were sequentially added to a solution of the D-(+)-trehalose dihydrate dissolved in anhydrous DMF (4 mL). The completion of the reaction was reached by stirring the mixture overnight at room temperature. The mixture was then diluted in MeOH (60 mL) before the addition of K_2_CO_3_ (0.11 equiv., 0.020 g, 0.145 mmol). It was then stirred 4 h at 0 °C, before a quenching of the reaction by the addition of NaHCO_3_ (50 mL, aq.) and a dilution with water (30 mL). The aqueous phase was extracted 3 times with EtOAc (3 × 30 mL), prior to a drying over Na_2_SO_4_ of the combined organic phases, and a concentration to dryness under vacuum. The expected compound **7** was obtained as a white powder by a purification using flash chromatography on a silica gel (petroleum ether/EtOAc/NEt_3_, 80:19:1) (0.530 g, 52%). **R_f_:** 0.42 (petroleum ether/EtOAc/NEt_3_, 80:19:1). **^1^H NMR** (300 MHz, MeOD) δ (ppm): 0.12–0.20 (m, 54H, 6 TMS), 1.90 (bs, 2H, OH), 3.48 (dd, 2H, *J* = 3.0 Hz, 9.3 Hz, H_2_, H_2′_), 3.49–3.52 (dd, 2H, H_4_, H_4′_), 3.65–3.71 (m, 4H, H_6_, H_6′_), 3.83–3.93 (m, 4H, H_3_, H_3′_, H_5_, H_5′_), 4.96 (d, 2H, H_1_, H_1′_). **^13^C NMR** (75.46 MHz, MeOD) δ (ppm): 0.3, 1.2, 1.6 (6 TMS), 61.7 (C_6_, C_6′_), 72.5, 74.4, 74.9 (C_2_, C_2′_, C_3_, C_3′_, C_4_, C_4′_), 74.0 (C_5_, C_5′_), 95.5 (C_1_, C_1′_).

### 2.9. Synthesis of Compound ***8***: 2,2′,3,3′,4,4′,6′-Hepta-O-(trimethylsilyl)-a,a’-D-trehalos-6-yl 3-[3′-O-(tert-butyldimethylsilyl)-5′-{4-[(dodecylcarbamoyloxy)methyl]-1H-1,2,3-triazol-1-yl}-5′-deoxythymidin-3-yl] Propanoate

Compound **7** (1 equiv., 0.126 g, 0.16 mmol), DCC (*N,N*′-Dicyclohexylcarbodiimide) (1.5 equiv., 0.050 g, 0.24 mmol) and DMAP (4-Dimethylaminopyridine) (0.5 equiv., 0.010 g, 0.08 mmol) were sequentially added to a solution of **6** (1.5 equiv., 0.175 g, 0.24 mmol) in anhydrous DCM (3 mL) and under argon. The mixture was stirred overnight at room temperature until the completion of the reaction. The precipitate was filtered and washed with DCM, followed by concentration to dryness of the filtrate under vacuum. The resulting crude was purified by flash chromatography on a silica gel (petroleum ether/EtOAc, 75:25) to obtain the expected compound **8** as a white powder (0.147 mg, 41%). **R_f_:** 0.49 (petroleum ether/EtOAc, 75:25). **^1^H NMR** (300 MHz, CDCl_3_) δ (ppm): −0.01–0.21 (m, 60H, CH_3_-Si et OTMS), 0.78–0.93 (m, 12H, CH_3_ lipid et ^t^Bu), 1.15–1.32 (m, 18H, CH_2_ lipid), 1.36–1.51 (m, 2H, CH_2_-CH_3_ lipid), 1.87 (s, 3H, CH_3_ thymidine), 1.96–2.13 (m, 1H, H_2′_), 2.17–2.31 (m, 1H, H_2′_), 2.60–2.75 (m, 2H, H_B_ linker), 3.04–3.16 (m, 2H, CH_2_-NH lipid), 3.34–3.51 (m, 4H, H_A_ linker et H_2_ trehalose, H_2′_ trehalose), 3.60–3.74 (m, 2H, H_6′_ trehalose), 3.77–4.33 (m, 10H, H_4′_, H_6_ trehalose, H_3_ trehalose, H_3′_ trehalose, H_4_ trehalose, H_4′_ trehalose_,_ H_5_ trehalose, H_5′_ trehalose et OH trehalose), 4.39–4.47 (m, 1H, H_3′_), 4.55–4.61 (m, 2H, H_5′_), 4.83–4.91 (d, 2H, H_1_ trehalose, H_1′_ trehalose), 4.93–5.02 (m, 1H, NH lipid), 5.13 (s, 2H, O-CH_2_ triazole), 6.19 (t, *J* = 6.6 Hz, 1H, H_1′_), 6.63 (s, 1H, H_6_), 7.69 (s, 1H, CH triazole). **^13^C NMR** (75.46 MHz, CDCl_3_) δ (ppm): −4.7 (CH_3_-Si), −4.5 (CH_3_-Si), 0.5 (OTMS), 1.2 (OTMS), 1.5 (OTMS), 13.3 (CH_3_ thymidine), 14.5 (CH_3_ lipid), 18.6 (Cq ^t^Bu), 23.6 (CH_2_ lipid), 25.9 (CH_3_ ^t^Bu), 26.2 (CH_2_ lipid), 26.6 (CH_2_ lipid), 27.7 (CH_2_ lipid), 30.2 (CH_2_ lipid), 30.3 (CH_2_ lipid), 30.6 (CH_2_ lipid), 30.7 (CH_2_ lipid), 32.9 (CH_2_-CH_3_ lipid), 33.1 (CH_2_-B linker), 34.6 (CH_2_ lipid), 38.2 (CH_2_-A linker), 40.2 (C_2′_), 41.7 (CH_2_-NH lipid), 52.0 (C_5′_), 58.2 (O-CH_2_ triazole), 61.6 (C_6′_ trehalose), 64.9 (C_6_ trehalose), 71.6, 72.4 (C_2_ trehalose, C_2′_ trehalose), 73.0 (C_5′_ trehalose), 73.5 (C_3′_), 73.7 (C_5_ trehalose), 73.8, 74.4, 74.6, 74.7 (C_3_ trehalose, C_3′_ trehalose, C_4_ trehalose, C_4′_ trehalose), 85.9 (C_4′_), 88.2 (C_1′_), 95.2, 95.4 (C_1_ trehalose_,_ C_1′_ trehalose), 111.0 (C_5_), 126.7 (CH triazole), 136.5 (C_6_), 144.7 (C triazole), 151.6 (C=O thymidine), 158.1 (C=O lipid), 164.8 (C=O thymidine), 172.6 (C=O linker). **HRMS** (ESI) [M + H^+^]: calcd = 1477.77427, found = 1477.77674.

### 2.10. Synthesis of Compound ***9*** (GNL): 5′-{4-[(Dodecylcarbamoyloxy)methyl]-1H-1,2,3-triazol-1-yl}-5′-deoxythymidin-3-yl Propanoate

Dowex 50WX2-100 ion exchange resin (0.350 g) was added to a solution of **8** (1 equiv., 0.105 g, 0.07 mmol) in MeOH (20 mL). The mixture was stirred for 15 min at room temperature until the completion of the reaction. The mixture was then filtered and the filtrate was evaporated under vacuum. Purification of compound **9**′ (0.062 g, 85%) was not necessary and it was directly engaged in the next reaction after drying. TBAF (1 equiv., 62 µL, 0.06 mmol) was added to a solution of **9**′ (1 equiv., 0.062 g, 0.06 mmol) in anhydrous THF (3 mL) and under argon at 0 °C. The mixture was stirred for 2 h at 0 °C until the completion of the reaction. The reaction was then quenched by the addition of water (5 mL, aq.). The expected compound **9** (white powder) was obtained after three extractions of the aqueous phase with DCM (3 × 10 mL), drying of the combined organic phases over Na_2_SO_4,_ a concentration to dryness under vacuum and a purification of the resulting crude by flash chromatography on a silica gel (DCM/MeOH/Water, 80:18.5:1.5) (0.029 g, 52%). **R_f_:** 0.31 (petroleum ether/EtOAc/NEt_3_, 80:18.5:1.5). **^1^H NMR** (300 MHz, MeOD) δ (ppm): 0.84–0.98 (m, 3H, CH_3_ lipid), 1.18–1.40 (m, 18H, CH_2_ lipid), 1.41–1.55 (m, 2H, CH_2_-CH_3_ lipid), 1.92 (s, 3H, CH_3_ thymidine), 2.21–2.39 (m, 2H, H_2′_), 2.65 (t, *J* = 7.2 Hz, 2H, H_B_ linker), 2.99–3.09 (m, 2H, CH_2_-NH lipid), 3.25–3.37 (m, 6H, H_2_ trehalose, H_2′_ trehalose_,_ H_3_ trehalose, H_3′_ trehalose, H_4_ trehalose et H_4′_ trehalose_,_), 3.45 (q, *J* = 3.9 Hz, 9.9 Hz, 2H, H_6′_ trehalose), 3.70–3.80 (m, 1H, H_5′_ trehalose), 3.97–4.05 (m, 1H, H_5_ trehalose), 4.11–4.36 (m, 5H, H_4′_, H_A_ linker, H_6_ trehalose), 4.36–4.44 (m, 1H, H_3′_), 4.65–4.79 (m, 2H, H_5′_), 4.83–5.02 (d, 2H, H_1_ trehalose_,_ H_1′_ trehalose), 5.10 (s, 2H, O-CH_2_ triazole), 6.15 (t, *J* = 6.9 Hz, 1H, H_1′_), 7.23 (s, 1H, H_6_), 8.00 (s, 1H, CH triazole). **^13^C NMR** (75.46 MHz, MeOD) δ (ppm): 13.1 (CH_3_ thymidine), 14.3 (CH_3_ lipid), 23.2 (CH_2_ lipid), 25.4 (CH_3_ ^t^Bu), 29.6 (CH_2_ lipid), 29.6 (CH_2_ lipid), 29.9 (CH_2_ lipid), 30.0 (CH_2_ lipid), 30.2 (CH_2_ lipid), 30.2 (CH_2_ lipid), 30.2 (CH_2_ lipid), 32.4 (CH_2_-CH_3_ lipid), 32.8 (CH_2_-B linker), 34.6 (CH_2_ lipid), 34.8 (CH_2_-A linker), 37.8 (C_2′_), 39.4 (CH_2_-NH lipid), 51.9 (C_5′_), 62.2 (O-CH_2_ triazole), 63.2 (C_6′_ trehalose), 64.9 (C_6_ trehalose), 69.2, 70.6 (C_2_ trehalose, C_2′_ trehalose), 70.9 (C_5′_ trehalose), 71.2 (C_3′_), 71.5 (C_5_ trehalose), 72.3, 73.0, 73.6, 73.8 (C_3_ trehalose, C_3′_ trehalose, C_4_ trehalose, C_4′_ trehalose), 84.6 (C_4′_), 87.3 (C_1′_), 94.3, 95.5 (C_1_ trehalose_,_ C_1′_ trehalose), 110.8 (C_5_), 125.3 (CH triazole), 136.9 (C_6_), 145.1 (C triazole), 151.2 (C=O thymidine), 155.6 (C=O lipid), 164.2 (C=O thymidine), 172.3 (C=O linker). **HRMS** (ESI) [M + H^+^]: calcd = 931.45356, found = 931.64586.

### 2.11. Synthesis of Compound ***10*** (dATr): 6,6′-Bis-O-dodecanoyl-α,α’-D-trehalose

Compound **7** (1 equiv., 0.200 g, 0.26 mmol), DCC (2 equiv., 0.106 g, 0.52 mmol) and DMAP (0.5 equiv., 0.016 g, 0.13 mmol) were sequentially added to a solution of lauric acid (2 equiv., 0.104 g, 0.52 mmol) in anhydrous DCM (3 mL) and under argon. The mixture was stirred overnight at room temperature until the completion of the reaction. The precipitate was filtered and washed with DCM, followed by concentration to dryness of the filtrate under vacuum. The resulting crude was purified by flash chromatography on a silica gel (petroleum ether/EtOAc, 75:25) to obtain the expected compound **7**′ as a white powder (0.254 mg, 86%). Next, Dowex 50WX2-100 ion exchange resin (0.700 g) was added to a solution of **8**′ (1 equiv., 0.254 g, 0.22 mmol) in MeOH (40 mL). The mixture was stirred for 15 min at room temperature until the completion of the reaction. The mixture was then filtered and the filtrate was evaporated under vacuum. Purification of compound **10** (0.143 g, 91%) was not necessary and it was directly engaged in the next reaction after drying. **R_f_:** 0.49 (petroleum ether/EtOAc, 75:25). **^1^H NMR** (300 MHz, MeOD) δ (ppm): 0.70–0.82 (m, 3H, CH_3_ lipid), 1.03–1.25 (m, 32H, CH_2_ lipid), 1.28–1.42 (m, 4H, CH_2_-CH_3_ lipid), 3.15–3.26 (m, 4H, CH_2_-C=O lipid), 3.50 (dd, 2H, *J* = 3.1 Hz, 9.2 Hz, H_2_, H_2′_), 3.49–3.58 (m, 2H, H_4_, H_4′_), 3.63–3.70 (m, 4H, H_6_, H_6′_), 3.83–3.90 (m, 2H, H_3_, H_3′_), 3.92–4.02 (m, 2H, H_5_, H_5′_), 4.96 (d, 2H, H_1_, H_1′_). **^13^C NMR** (75.46 MHz, MeOD) δ (ppm): 13.8 (CH_3_ lipid), 20.5 (CH_2_ lipid), 24.7 (CH_2_ lipid), 30.1 (CH_2_ lipid), 30.5 (CH_2_ lipid), 30.6 (CH_2_ lipid), 30.6 (CH_2_ lipid), 30.9 (CH_2_ lipid), 30.9 (CH_2_-CH_3_ lipid), 31.1 (CH_2_ lipid), 31.5 (CH_2_ lipid), 61.7, 61.8 (C_6_, C_6′_), 71.3, 72.5, 74.4 (C_2_, C_2′_, C_3_, C_3′_, C_4_, C_4′_), 74.9 (C_5_, C_5′_), 95.5 (C_1_, C_1′_).

### 2.12. General Procedure for Nanoparticles and Solid Lipid Nanoparticles Preparation

The PLGA was purchased from Sigma-Aldrich (Resomer^®^ RG 503H PLGA; lactide-glycoside ratio of 50:50; molecular weight from 24,000 to 38,000). The nanoprecipitation protocol was adapted from a previous study to prepare NP suspensions [19,34]. Briefly, 31 mg of Resomer^®^ RG 503H PLGA was dissolved in 3.1 mL of tetrahydrofuran (THF; Sigma-Aldrich). Under sonication (37Hz, 100%, 3 min), 200 µL of the previous solution was quickly added to 20 mL of deionized water. The resulting NP suspension was then slowly concentrated by centrifugation using 10 kDa centrifugal filters (Merck Millipore, Saint Quentin Fallavier, France, Amicon Ultra centrifugal Filter) for 21 min at 5000 RPM (Centrifuge 5804 R, Eppendorf). Once the supernatant had been collected, the resulting suspension was ready for use (typical concentration of NPs ~ 0.2 mg·mL^−1^). A rapid mixing protocol was adapted from the literature to prepare the NP suspension. 500 μL of the previously prepared stock solution was simultaneously injected (3 mL·min^−1^) with 1.5 mL of deionized water (9 mL·min^−1^) using syringe pumps (kdScientific, Holliston, MA, USA). The obtained PLGA NP suspension was concentrated to a typical concentration of 2.5 mg·mL^−1^. For the control condition, PLGA NPs loaded with Nile Red fluorophore (Sigma-Aldrich, St. Louis, MO, USA 19123) were formulated as previously described, except that 0.9 mg of Nile Red was added to the polymer stock solution. **F2** and **F3** were formulated similarly, except that 15% (*w*/*w*) of **dATr** or **GNL** were introduced into the stock solution (typical final concentration in trehalose 0.4 µM). For all experiments, PLGA NP suspensions were prepared extemporaneously.

### 2.13. Nanoparticle and Solid Lipid Nanoparticle Characterization and Stability Evaluation

Granulometric profile of NPs and SLNs were acquired by Dynamic Light Scattering (DLS), using Malvern Instruments (Zetasizer Nano ZS, Palaiseau, France). NPs/SLNs were diluted at 1:1000 (*v*/*v*), then the average size and the polydispersity index were obtained in triplicate at 25 °C. To determine the Zeta potential, NPs/SLNs, at the same dilution, were analyzed using a Zetasizer Nano ZS coupled with Folded Capillary Cell (DTS1060). Short-term stability assessment was performed for a period of one month by visually checking the lack of creaming or phase separation and monitoring the size and zeta potential. Transmission electron microscopy was carried out using a Hitachi (Tokyo, Japan) coupled to an ORIUS SC1000 11MPX (Gatan, Pleasanton, CA, USA). All samples were deposited on carbon-coated grids (Delta Microscopies, Mauressac, France) and dried after 3 min of contact. A staining procedure using uranyless (Delta Microscopies, Mauressac, France) or nanotungstene was used.

### 2.14. Quantification of the Free GNL by Enzymatic Assay

Quantification of the free GNL in the suspensions of trehalose derivative-loaded NPs was performed using the Megazyme microplate assay procedure (Trehalose Assay Kit, Megazyne, Bray, Ireland) [34]. Briefly, trehalose standard solutions (concentration between 25 and 100 µM) were prepared and 20 µL of each was used for the calibration curve. To these 20 µL were added 200 µL of deionized water, 20 µL of solution **1** (buffer), 10 µL of solution **2** (containing both nicotinamide adenine dinucleotide phosphate (NADP^+^) and adenosine 5′ triphosphate) and 2 µL of solution **3** (containing the hexokinase and glucose-6-phosphate dehydrogenase). The solutions were mixed and the absorbance was measured at 340 nm (abs_0_) before addition to the trehalase medium. The next step was the addition of 2 µL of suspension **4**, containing trehalase, to the wells. The 96-well plate was shaken and incubated for 8 min at room temperature before the absorbance of the different solutions at 340 nm (abs_1_) was read. The calibration curve was thus obtained by representing the background-corrected absorbance (abs_1_-abs_0_) as a function of the trehalose concentration. For the quantification of the NP samples, the same protocol was used except for the direct use of 220 µL of suspension instead of 20 µL of standard and 200 µL of deionized water, to ensure absorbances included in the calibration range.

### 2.15. Encapsulation Efficiency and Drug Loading

For the determination of the encapsulation efficiency (EE) and the drug loading (DL) of PLGA NPs, **GNL** quantification was achieved by UV-Vis spectroscopy (Jasco V-630, France) using a standard calibration curve of **GNL** dissolved in methanol (between 5 and 40 µg·mL^−1^; absorbance at 266.5 nm).Equation (1) was used, where y is the absorbance value and x the **GNL** concentration. The standard calibration curve was linear over the range of 5–40 µg·mL^−1^ with r^2^ = 0.9993. The amount of encapsulated **GNL** in the PLGA NPs (also called DL) was calculated from the mass of the incorporated drug, using Equation (2):y = 0.0088x + 0.1572(1)
(2)$$\mathrm{DL}=\frac{\mathrm{Amount}\;\mathrm{of}\;GNL\;\mathrm{in}\;\mathrm{NPs}}{\mathrm{Amount}\;\mathrm{of}\;GNL\;\mathrm{loaded}\;\mathrm{NPs}}\times 100$$

The drug EE was defined as the ratio of the mass of the encapsulated drug over the mass of the drug used for the formulation, using the following equation:(3)$$\mathrm{EE}=\frac{\mathrm{Amount}\;\mathrm{of}\;\mathrm{encapsulated}\;GNL}{\mathrm{Amount}\;\mathrm{of}\;GNL\;\mathrm{used}\;\mathrm{for}\;\mathrm{NP}\;\mathrm{formulation}}\times 100$$

The drug EE and the DL efficiency were calculated using Equations (1)–(3).

### 2.16. Infrared Analyses

Spectra of **PLGA**, **GNL**, **F1** and **F3** were acquired using an FT-IR spectrometer (FT-IR 4600, Jasco Inc., Tokyo, Japan) controlled by spectra manager software. Samples (3 mg) were mixed with 300 mg of KBr powder and then compressed with a mechanical die press to obtain translucent pellets. All spectra were recorded after averaging 16 accumulations with 8 cm^−1^ resolution between 4000 cm^−1^ and 450 cm^−1^. Background was performed on the atmosphere and subtracted to all acquisitions. All spectra were acquired in the transmittance mode. IR data were processed using spectra manager for IR spectra and IR derivatives (Savitzky–Golay: derivative order 1, polynome order 2 and window of 7 points).

### 2.17. Cell Culture and Cell Viability Assay

Human neuroblastoma cell lines (BE(2)-M17) from ATCC (CRL-2267) were grown in Opti-MEM (Life Technologies, Carlsbad, CA, USA; 31985-047) supplemented with 10% fetal bovine serum (Sigma-Aldrich) and 1% penicillin/streptomycin. All NP and SLN suspensions were used as freshly prepared on cells grown at 70% to 80% confluence. Cells were exposed at 0.5 µL of loaded NPs and SLNs for 24 h or 48 h, and each experiment was performed at least in triplicate. MTS assay (ATCC/LGC Promochem, Molsheim, France) estimated cell viability using the manufacturer instructions.

### 2.18. Western Blot Analysis

BE(2)-M17 cells were seeded on 6-well plates (NUNC) and grown to 70–80% confluence per well before being treated with **F1**, **F2**, **F3**, **F4** and molecular trehalose as control (concentration of 0.4 µM). The cells were then transferred to Eppendorf tubes before centrifugation for 5 min at 3000 rpm and 4 °C. The supernatant was removed before the dropwise addition of 50 µL of P/X solution (P/X = 1.5 mL RB1X, 0.450 mL P4X, 0.050 mL dithiothreitol (DTT)) under mechanical agitation. The samples were then incubated for 5 min at 100 °C. Protein samples were loaded into 18% acrylamide gels and separated by SDS-PAGE (sodium dodecyl sulphate-polyacrylamide gel electrophoresis) before transfer to a nitrocellulose membrane (0.2 µm—Biorad—USA). The membranes were then blocked in 5% PBS (phosphate-buffered saline)-milk buffer before overnight incubation at 4 °C with rabbit anti-LC3 (light chain 3) primary antibody (1/1000°, Novus Biological, Littleton, CO, USA #NB100-2220). An incubation with mouse anti-actin antibody (1/5000°, Sigma) served as a loading control. The appropriate secondary antibodies were then used before revelation with the revelation kit (Super Signal West Pico Chemiluminescent kit—Immobilon Western, Chemiluminescent HRP substrate, Millipore). Chemiluminescence images were acquired using a ChemiDoc + XRS analysis system (BioRad, Hercules, CA, USA). The signal per well was quantified using ImageJ software and normalized by actin before statistical analysis.

### 2.19. Immunostaining and Imaging

Extemporaneous NP suspensions were used for all experiments. In the case of Nile Red-loaded NPs internalization and colocalization imaging assays, BE(2)-M17 cells were replated in 6-well plates (NUNC) with coverslips after trypsinization, and allowed growing at 70% to 80% confluence per well. They were then exposed to the tested formulations for 24 h at 37 °C. The final concentration of NPs in each well was set at ~3 µg·mL^−1^ for **F5**. The cell fixation was performed using 4% paraformaldehyde for 20 min at 4 °C, prior to the washing step with PBS 1X solution. A mixture, composed of 225 µL of triton, 450 µL of normal goat serum and 5 mL of PBS 1X, was added for cell permeabilization and blocking steps. The LAMP2 (Mx, H4B4) antibodies and their appropriate secondary antibodies conjugated with GAM 488 (Life Technologies) were used to mark the lysosomes overnight at 4 °C and for staining, respectively. The cells were incubated with 8 µM Hoechst dye (ThermoFisher Scientific, Waltham, MA, USA, #3342) (8 min; room temperature) to stain cytoplasm and nuclei, prior to mounting. After an air-drying step, slices were mounted on #1.5 coverslips using Dako fluorescent mounting medium, then left overnight in darkness to dry. A Leica TCS SP8 laser scanning confocal microscope (Leica Microsystems, Wetzlar, Germany) combined with a 63X Plan Apo CS oil immersion objective was used to acquire image stacks (pixel size ~100 nm; z-step 0.3 µm). Nile Red was observed in a detection window between 570 and 590 nm after an excitation at 568 nm (DPSS laser). For lysosomes, the excitation was performed at 488 nm (argon laser) and the detection within a window between 545 and 605 nm. In parallel, the untreated cells were required for autofluorescence controls using the previously described parameters. LAS AF v2.6 acquisition software implemented with HCS-A module (Leica Microsystems) was used for image analyses. The relative intensity of Nile Red to cell surface was determined using Definiens XD Developer v2.5 software (Definiens). The images were analyzed with the ImageJ (NIH, Bethesda, MD, USA; Fiji) software for colocalization, fluorescence profiles, orthogonal projections and maximal intensity projections.

### 2.20. GFP-LC3 Reporter Autophagy Assay

To assess autophagy activity, the green fluorescent protein (GFP)-LC3 reporter was used as previously described [29,35]. Briefly, a BE(2)-M17 cell line was plated on coverslips in 12-well plates which were transfected with mCherry-GFP-LC3 plasmid at 1.6 µg DNA using polyethylenimine (PEI)-mediated transfection. Cells were maintained for 24 h at 37 °C in 5% CO_2_ before being treated with unloaded or loaded NPs or SLNs for 24 h at 1:1000. Cell coverslips were fixed at 4 °C for 20 min using 4% paraformaldehyde, followed by 3 washing steps of 5 min with PBS 1X. The final step consisted in a staining using DAPI solution (10 μM; Invitrogen, Waltham, MA, USA) for 8 min prior to long washes in PBS 1X solution. Mounting solution (Dako) was used to mount the coverslips onto slides, and images were obtained using a wide-field Olympus Epifluorescent Microscope (BX3-CBH) coupled with a Hamamatsu camera (ORCA-Flash 4.0 LT). Images were deconvolved using the cellSens Dimension software. Image analysis was performed using Fiji/ImageJ software.

### 2.21. Statistical Analysis

Statistical analyses were performed using GraphPad Prism 6 software. In the case of functional assays, a one-way analysis of variance (ANOVA) was used to evaluate the data statistical significance prior to a Tukey’s multiple comparison test. A significant difference was highlighted by *p* < 0.05. Data appear as estimation graphics called ‘Gardner–Altman plots’: in the top graph, the data from **UT** (untreated), **F1**, **F2**, **F3** and **F4** groups are presented as scatter plots showing the observed values, along with the above-defined descriptive statistics (mean ± standard deviation). Below each graph, a contrast graph using the difference axis to display an effect size, here the mean difference. Vertically aligned with the mean of the UT group, the mean difference is indicated by the black circle. The 95% confidence interval (CI) of the mean difference is illustrated by the black vertical line. The curve indicates the resampled distribution of the effect size, given the observed data.

## 3. Results and Discussion

### 3.1. Design and Synthesis of Glyconucleolipidic Platforms

Two original compounds, a diacyl trehalose derivative (**dATr**, compound **10**) and a glyconucleolipid (**GNL**, compound **9**) composed of a single C12 lipid chain featuring trehalose as the carbohydrate moiety, were designed and synthesized (Figure 1). **GNL** was synthesized by a ten-step synthesis route beginning with the functionalization of commercially available thymidine. In the first step, an azide group was introduced at the 5′ position of the nucleoside through an Appel reaction, followed by a nucleophilic displacement of the bromine atom with NaN_3_ to form the azidothymidine **1**. The linker, coupling the NL moiety to trehalose, was added by a Michael reaction to form **2**. After protecting the 3′ position of thymidine, **3** was submitted to a CuAAC (Copper(I)-catalysed alkyne-azide cycloaddition) reaction with the previously prepared propargylated lipid chain **4** to give the corresponding 1,2,3-triazole derivative **5**. The latter was saponified in basic medium, followed by esterification with the per-silylated trehalose **7,** leading to compound **8**. NLs were coupled to trehalose through an ester moiety, allowing its enzymatic release in neuronal cells once transportation through the lipid bilayers had been achieved. Deprotection of the silyl groups under acidic conditions yielded the final **GNL** structure. Compound **10** (**dATr**) was obtained by a selective protection of trehalose dihydrate **7**, first followed by an esterification step to append the lauric chain and then followed by a deprotection step by means of Dowex 50WX2-100 ion exchange resin.

### 3.2. Physico-Chemical Characterization of PLGA Nanoparticles and Solid Lipid Nanoparticles

To evaluate either the added value to link trehalose to a NL or the PLGA NPs as a nanocarrier, five formulations were prepared. First, we used unloaded PLGA NPs (**F1**) as a control. Second, **dATr** was formulated into PLGA NPs, named **dATr**-loaded PLGA NPs (**F2**) and **GNL** was also formulated into PLGA NPs, described as **GNL**-loaded PLGA NPs (**F3**). By comparing **F2** and **F3**, we were able to gain insight into the contribution of NLs. Third, to assess whether a PLGA-free formulation was appropriate as a PLGA alternative, we took advantage of the self-assembling properties of **GNLs** to formulate them into SLNs [16,17] (**F4**). Finally, Nile Red-loaded PLGA NPs (**F5**) were prepared as an experimental readout to evaluate exclusively the internalization into neuronal cells (Figure 1).

These nanovectors were first obtained by nanoprecipitation, and then by rapid mixing. Nanoprecipitation requires the addition of two miscible solvents, in this case THF and water, and results in the spontaneous formation of NPs on phase separation. With dissolved PLGA and active compounds (for loaded NPs), the organic phase was added dropwise to the aqueous medium under sonication. Once the nanocarriers were formed, the particle dispersions were further processed to purify and concentrate them through THF rotary evaporation and dialysis. The main drawback of this method was the loss of product due to the aggregation of PLGA during the process, and therefore, the low yield of production. On the other hand, rapid mixing involves the simultaneous injection of an organic phase (THF) and the aqueous medium using a syringe pump followed by a T-junction mixing [36]. The self-assembly of smaller NPs is thus produced by a supersaturation of PLGA and/or **GNL** in the medium, induced by the rapid increase in the polarity of the two miscible phases [36]. Compared to nanoprecipitation, rapid mixing has multiple advantages, such as quicker formulation of smaller nano-objects, allowing them to be obtained in 15 min instead of 45 min in the case of nanoprecipitation, and at a higher concentration (0.2 mg·mL^−1^ of NPs using nanoprecipitation versus 2.5 mg·mL^−1^ using rapid mixing). Both trehalose derivatives were successfully incorporated into PLGA NPs or able to form SLNs. Therefore, three different formulations (**F1**, **F2**, **F3**) were prepared by both nanoprecipitation and rapid mixing to observe the process impact on the physico-chemical characteristics (size, polydispersity index (monodisperse suspension under 0.2) and zeta potential) (Table 1).

A significant decrease (student’s test, *p* < 0.05) in the mean diameter was observed for the three formulations (**F1**/**F2**/**F3**) using rapid mixing after dialysis. NPs with particle sizes below 200 nm were obtained, allowing membrane and biological barrier passage, as well as an increase in the half-life, crucial parameters for biomedical applications [37,38]. Another advantage of this approach was obtaining new nano-objects only made of **GNL**, i.e., SLN. These were formulated following the same protocol as the PLGA NPs, except for the polymer addition phase. These SLNs with a diameter close to PLGA-based NPs (120.4 ± 1.4 nm vs. 167.2 ± 2.4 nm) had a surface charge of −16.2 ± 1.7 mV. The colloidal stability of the formulations obtained by rapid mixing was evaluated at 4 °C (storage condition) and 37 °C (use condition). In terms of diameter (Figure 2A), the objects were stable for at least 20 days at 4 °C, whereas at 37 °C, a destabilization, characterized by an increase in the diameter (10-fold), polydispersity (Figure 2B) and a modification of the zeta potential (Figure 2C), were observed from 10 days. Interestingly enough, the size of SLNs remained stable despite low surface loading (Figure 2A,C). This tendency may be due to the particular structure of SLNs based on physical interactions. Moreover, the in vitro biological analyses were carried out only during 3 days, ensuring the object stability.

To dig deeper into the characterization of these nano-objects, two techniques were used: transmission electron microscopy (TEM) and infrared (IR) spectroscopy. The first one confirmed the size of the particles and their spherical shape (Figure 2D). The advantage of this study was obvious in the case of **F5**. Indeed, the study by DLS of this formulation containing Nile Red was not possible owing to the similarity in color between the chromophore and the laser used. TEM was, therefore, able to overcome this limit, allowing the observation of spherical objects with an average diameter of 118.8 ± 25.8 nm consistent with the results obtained for other PLGA NPs. The second confirmed the presence of **GNL** in the NPs. From the results of IR studies, specific bands for each compound (**PLGA** and **GNL**) were identified (Table 2). These specific vibrations were also observable for the nano-objects, highlighting the presence of both **PLGA** and **GNL** (Appendix A).

Due to the low proportion of **GNL** in the NPs, the band intensity was low. The ν_N = N_ band was overlaid by other signals (ν_C = C_), leading to the observation of a shoulder. Signal deconvolusion using a Savitzky–Golay derivative was employed to separate signals from each pure compound (**PLGA** and **GNL**). Focusing on the **GNL** molecule and based on its first derivative spectrum, five specific signals (C=O, N=N, and C-O elongation vibrations, Table 2) were identified for **GNL**-loaded PLGA NPs. This result corroborates those obtained with the IR spectra and increased the number of signals related to the active compound. It confirms the presence of **GNL** in the PLGA NPs, even if at low concentration.

### 3.3. GNL Quantification in PLGA Nanoparticles

Two quantification approaches were used to (i) quantify the free (or non-encapsulated) **GNL** and **dATr** and (ii) quantify the encapsulated **GNL**. The strategy for the free **GNL** was based on an enzymatic assay with UV quantification [39]. Indeed, the use of trehalase allowed the splitting of the trehalose into two glucose units which, after enzymatic reactions, enabled a detection at 340 nm. In the case of PLGA NP loaded with **GNL** or **dATr**, the concentrations of free active substances were found equivalent to zero, confirming their encapsulation and similar behavior (Appendix A). Regarding the DL and the drug EE, a destabilization of the nano-objects in basic conditions allowed the quantification of **GNL** by UV-Vis spectroscopy at 266.5 nm. A DL and an EE of 15% and 96%, respectively, were determined for **F3** and agreed with the values usually found in the literature (Appendix A) [36].

### 3.4. Biocompatibility and Internalization of GNL Formulations

**F2**, **F3**, and **F4** cytotoxicity at low concentration (1/1000°) was assessed in vitro on human neuroblastoma cell lines (i.e., BE(2)-M17 cells), given that previous studies showed that **F1** and **F5** were biocompatible at this concentration [19,20,21,22]. After an exposure of 24 h and 48 h, the tested formulations (**F2**, **F3**, and **F4**) did not exhibit significant toxicity compared to untreated cells (Figure 3A,B).

The next step corresponded to the evaluation of the cell internalization. As previously mentioned, a dedicated formulation containing Nile Red was prepared (**F5**) to evaluate cellular uptake. BE(2)-M17 cells (healthy neurons) were incubated in the presence of **F5** diluted 1000 times for 24 h. As previously described [19,20,21], PLGA NPs were internalized by cells (Figure 3C). Overall, the results obtained with the different formulations indicate their safety and cellular uptake at low concentration.

### 3.5. GNL-Loaded PLGA Nanoparticle Activity as Autophagy Inducer

The biological effects of the different formulations were further explored to monitor autophagic activity, following levels of the microtubule-associated protein, LC3, a key protein in the autophagy pathway, in the BE(2)-M17 healthy neuroblastoma cell line [30]. Trehalose-derivate nano-objects (**F2**, **F3**, **F4**, with the same concentration in trehalose) exhibited by immunoblot an accumulation of autophagosomes (APs) after exposure, compared with untreated cells, as indicated by increased levels in the AP marker LC3B-II in the cells (Figure 4A).

In particular, **F2** resulted in a 1.5-fold increase, while **F3** and **F4** significantly increased up to two-fold compared with UT conditions and molecular trehalose (used as control, Appendix A). These effects might be explained by the amphiphilic nature of NL and their resemblance to the lipid bilayer that composes cell membranes. NLs could cross the cell membrane more easily with or without the help of membrane transporters. These results indicate the beneficial effect of functionalizing the trehalose with a NL. The capacity to induce autophagy was further confirmed by immunofluorescence using a GFP-LC3 construct, which allowed the identification of AP-related vesicles. Further supporting an accumulation of AP in **F2**, **F3** and **F4**-treated cells, GFP-positive vesicles were markedly increased in these treated cells, indicating an increase in AP numbers (Figure 4B). Of note, it is important to keep in mind in this immunofluorescence assay that we can underestimate the number of GFP-positive vesicles due to the dynamic turnover of GFP-LC3 in the autolysosome (i.e., the pH sensitivity of GFP) [40]. Taken together, these data indicate that **GNL** significantly modulated the autophagy in these cells.

## 4. Conclusions

Herein, in order to develop new pharmaceutical nanotechnologies, the synthesis of a thymidine-derived **GNL**-bearing trehalose, as an active substance, to study their biocompatibility and ability to modulate autophagy for lysosomal impairment treatment was investigated. An amphiphilic molecule, named **GNL**, was designed and formulated into two different nanosystems, SLNs and PLGA NPs. These nano-objects were stable for at least 20 days at 4 °C and 10 days at 37 °C. The successful encapsulation of **GNL** inside PLGA NPs was confirmed by IR and UV-Visible spectroscopies (DL 15% and EE 96%). Cytotoxicity evaluation of **GNL**-loaded PLGA NPs and **GNL**-based SLN on human neuronal cells showed that **GNLs** are biocompatible at low concentration after 24 h of exposure. The uptake of fluorophore-loaded PLGA NPs into cells indicates that PLGA NPs are successfully internalized, highlighting the potential of these PLGA NPs as drug delivery systems. Moreover, immunoblotting and transfection assays suggest that **GNL**-based nanosystems improved biological activity of trehalose (two-fold) compared with molecular trehalose, thereby significantly modulating autophagy. These results suggest that **GNL**-based nanovectors can enhance the uptake of trehalose into cells and thus trehalose activity. Therefore, further in vitro studies will be carried out in a PD cellular model (M17 cells overexpressing α-synuclein or BE(2)-M17 cells with depletion of lysosomal type 5 P-type ATPase (ATP13A2)) and further characterization, in particular autophagy flux experiments associated with **GNL**-based nanosystems will be assessed. **GNL**-loaded functionalized NPs will be developed and in vivo assays will be performed to evaluate their ability to cross the BBB and target neurons of interest, notably in order to modulate the pH and lysosomal activity.

## Data Availability

The original contributions presented in the study are included in the article, further inquiries can be directed to the corresponding author.

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
