# Peer review of "Trehalose-Based Nucleolipids as Nanocarriers for Autophagy Modulation: An In Vitro Study"

_pharmaceutics, 2022, doi:10.3390/pharmaceutics14040857_

Round 1

Reviewer 1 Report

The reviewer's concerns have been addressed by authors and the concerns have been resolved in the current manuscript. Since the removal of descriptions that can be overstated and the changes to appropriate title, the title, research content, and conclusions are kept consistent.

The targeting system using trehalose seems to be very interesting and novel.

Reviewer 2 Report

Can be accepted at the current stage.

This manuscript is a resubmission of an earlier submission. The following is a list of the peer review reports and author responses from that submission.

Round 1

Reviewer 1 Report

The manuscript with a title „Trehalose-based Nucleolipids as efficient nanocarriers for authophagy modulation” by Cunha et al., describes the synthesis of novel nanoparticles and in vitro characterization of their biological activity. The important advantage of the described nanosystems is the use of trehalose, which is a small natural autophagy enhancer. The use of trehalose as an enhancer of autophagy may be useful for the treatment of neurodegenerative diseases with defects in the lysosomal autophagy pathway, as observed in Parkinson’s disease.

The manuscript is well written. The introduction provides a comprehensive overview of  relevant literature on  neurodegenerative diseases and  current strategies for drug delivery the into the central nervous system and for crossing the blood-brain barrier.

The synthesis of nanoformulations is described in detail under “Methods”. The in vitro experiments with a human neuroblastoma cell line are convincingly performed and presentd.  The results are well presented and the conclusion is based on the results.

I propose that this manuscript can be accepted as is.

Reviewer 2 Report

Cunha et al have been developed the trehalose and an amphiphilic nucleolipid conjugate nanoparticle and investigated the effects of the nanoparticle on the autophagy activity. the study that the nanoparticle synthesized with trehalose and glycol-nucleolipid can deliver to the neuronal cells and it has possible that the nano-system can be useful for the treatment of neurodegenerative disorders, was very interesting. Please consider the concerns listed below.

Line 50th to 52nd in Introduction, Trehalose has the property of widely distribution through the tissues of whole body. How can the formulation that developed in this study control the targeting to tissue (brain) in vivo? In this paper, the authors investigated the in vitro uptake study, but not study whether the formulation can delivery to the brain tissue in vivo.

Line 598th in Results, on the cytotoxicity, authors was described using previous reports (reference No. of 19 to 21) as a comparison. If there is already a report and this study was based on previous findings, is it worth discussing in this paper? Also, what is the difference between the formulations developed in this study and those reported in the previous study, and what is the novelty of this study?

In Conclusions, what is described in the conclusions seems to be far from the evidence based on the results evaluated in this paper. There are many future prospects and speculations, and the content described can be perceived as overexpression.

In 664th to 665th, why can the authors conclude from the results in this study that formulation is taken up to dopamine neuron? Will the nanoparticle prepared in this study actually pass through the BBB and be delivered into the dopamine neuron in vivo?

How are the release properties of trehalose? Is it the nanoparticle have the properties that trehalose release only after it has permeated BBB, and not release in blood or other tissue before transporting to the CNS regions?

Reviewer 3 Report

The authors identify that trehalose-based nucleolipids as nanocarriers can induce autophgy, which has potential therapeutic significance for human disorders. The research topic is interesting. However, it seems that there are major defects on experimental design and result deciphering.

1, the author only check LC3-II level and LC3-positive dots in cells, before drawing conclusion that the nanoparticles can induce autophagy. The increased level of LC3-II or an accumulation of LC3 puncta is not always indicative of autophagy induction and may represent a blockade in autophagosome maturation (J Biol Chem 2006; 281:36303–16). Therefore autophagic flux should be monitored before induction or inhibition of autophagy can be concluded. The term “autophagic flux” is used to represent the dynamic process of autophagy. In detail, autophagic flux refers to the whole process of autophagy, including autophagosome formation, maturation, fusion with lysosomes, subsequent break-down and the release of macromolecules back into the cytosol. The author should check p62 degradation and degradation of long-lived proteins, as well as observation of induced formation of autophagosome under electro-microscopy. Furthermore, autophagy inhibitor, such as 3-MA or Bafilomycin A1 should be added to consolidate their conclusions.

2, So far the roles of trahalose in modulation of autophagy are still under debates. In a recent study, it is found that treatment of trehalose results in accumulation of lipidated LC3 (LC3-II), p62, and autophagosomes, whereas it decreased autolysosomes in human neuroblastoma and primary rat cortical neurons. On the other hand, addition of Bafilomycin A1 to trehalose treatments had relatively marginal effect (Cell Death and Disease (2017) 8, e3091). In Figure 4A of the current study, the difference of LC3-II among groups are not significant. In Figure4B, F1 treated cells also have high level of LC3-II positive dots, which is not different from F2 and F3. Therefore it seems that nanoparticles, rather than trahalose residues may trigger the autophagy process. The authors should provide more evidence to explain the situation.

3, it is suggested that a positive autophagy inducer, such as rapamycin as well as single molecular trehalose should be added so as to observe and compare the exact effects of autophagy induced by nanoparticles conjugated with trehalose.